# Transcriptome Analysis of Glycerin Regulating Reuterin Production of *Lactobacillus reuteri*

**DOI:** 10.3390/microorganisms11082007

**Published:** 2023-08-04

**Authors:** Jingjing Wang, Qiang Yin, Han Bai, Wei Wang, Yajun Chen, Minghui Zhou, Ran Zhang, Guoao Ding, Zhongdong Xu, Yan Zhang

**Affiliations:** 1Department of Life Science, Hefei Normal University, Hefei 230061, China; wjj_0203@126.com (J.W.); bh010629@163.com (H.B.); wangwei_sir@126.com (W.W.); 18225870205@163.com (Y.C.); iandfake342623@126.com (M.Z.); d22301088@stu.ahu.edu.cn (R.Z.); d22301130@stu.ahu.edu.cn (G.D.); xuzhongdong@hfnu.edu.cn (Z.X.); 2Agricultural Engineering Research Institute, Anhui Academy of Agricultural Sciences, Nongke South Road 40, Hefei 230001, China; yinq301@126.com

**Keywords:** bacteriostasis, *Lactobacillus reuteri*, 3-hydroxypropannaldehyde, transcriptome

## Abstract

Reuterin can be produced from glycerol dehydration catalyzed by glycerol dehydratase (GDHt) in *Lactobacillus reuteri* and has broad application prospects in industry, agriculture, food, and other fields as it is active against prokaryotic and eukaryotic organisms and is resistant to proteases and lipases. However, high concentrations of glycerin inhibit reuterin production, and the mechanism behind this phenomenon is not clear. To elucidate the inhibitory mechanism of glycerol on reuterin synthesis in *L. reuteri* and provide reference data for constructing an *L. reuteri* culture system for highly effective 3-hydroxypropionaldehyde synthesis, we used transcriptome-sequencing technology to compare the morphologies and transcriptomes of *L. reuteri* cultured in a medium with or without 600 mM of glycerol. Our results showed that after the addition of 600 mM of glycerol to the culture medium and incubation for 10 h at 37 °C, the culture medium of *L. reuteri* LR301 exhibited the best bacteriostatic effect, and the morphology of *L. reuteri* cells had significantly changed. The addition of 600 mM of glycerol to the culture medium significantly altered the transcriptome and significantly downregulated the transcription of genes involved in glycol metabolism, such as *gldA*, *dhaT*, *glpK*, *plsX*, and *plsY*, but significantly upregulated the transcription of genes related to D-glucose synthesis.

## 1. Introduction

*Lactobacillus reuteri* is a probiotic with broad-spectrum antibacterial activity. It not only has the characteristics of lactic acid bacteria that are able to secrete lactic acid and enzymes but also can produce reuterin, a broad-spectrum antibacterial substance with the following chemical composition: 3-hydroxypropionaldehyde (3-HPA) [1]. Due to its ability to effectively inhibit bacteria, yeasts, fungi, and pathogens, 3-HPA is widely used for antiseptic and disease treatments [2,3]. In general, 3-PHA is stored in vivo as an intermediate product during metabolism [4]. In contrast, 3-PHA produced by *L. reuteri* can be secreted outside the bacterial cells, leading to the stable accumulation of 3-HPA in vitro and the strong resistance of *L. reuteri* to 3-HPA.

The oxidation and reduction pathways are the two pathways of glycerol metabolism in *L. reuteri*. In the reduction pathway, glycerol dehydration is catalyzed by glycerol dehydratase (GDHt, EC4.2.1.30) to remove one molecule of water to form 3-HPA in the presence of the co-enzyme B_12_. The produced 3-HPA is then converted to 1,3-propanediol dehydrogenase, depending on the presence of NAD^+^. 3-HPA is the first metabolite of glycerol; therefore, the yield of 3-HPA is as high as that of GDHt and is accompanied by a high conversion rate of glycerol, which may reach 85%. GDHt acts as a rate-limiting enzyme and plays a key role in the formation of 3-HPA. Therefore, the properties of GDHt are closely related to the antibacterial performance of *L. reuteri* [5]. In particular, in vitro studies have revealed that glycerol can induce the suicidal inactivation of GDHt [6]. GDHt produced by *L. reuteri* catalyzes the production of 3-HPA in the presence of glycerol in vitro. In the aforementioned studies, when the reaction time reached 60 min, the production of 3-HPA did not increase, and GDHt showed no activity. However, enzyme activity was detected again after the addition of glycerol. In an antibacterial activity experiment involving intestinal pathogenic bacteria, *L. reuteri* from fermented milk was cultured in both a normal medium and a medium supplemented with 200 mM of glycerol. The former exhibited obvious inhibition zones, but the glycerol culture did not form inhibition zones, indicating that GDHt could be inhibited by glycerol, which limited 3-HPA synthesis.

There is a contradiction between the suicidal inactivation and catalytic function of GDHt in glycerol metabolism. This has become the biggest bottleneck for the antibacterial application of *L. reuteri*. Clarifying this regulatory mechanism might resolve the application limitations of *L. reuteri* and provide a reference for related research on substrate inhibition. The catalytic characteristics of GDHt can be divided into coenzyme B_12_-dependent and coenzyme B_12_-independent types; GDHt in *L. reuteri* belongs to the former type. Coenzyme B_12_ undergoes Co-C isolysis in the presence of glycerol to form alkylcobalamin analogs that bind to GDHt and inhibit its activity. With the assistance of ATP, Mg^2+^, reactivation factors, and coenzyme B_12_, GDHt can be reactivated in situ [7].

The inhibitory effect of glycerol on 3-HPA synthesis was explored at different concentrations. When 50 g/L of glycerol and 50.6 g/L of carbohydrazide were added to a culture medium of *L. reuteri*, the inactivation rate of GDHt was reduced through the in situ complexation reaction of aldehyde and carbohydrazide. The yield of 3-HPA was 1.9 times higher than that of normal cultures [8]. The genes encoding GDHt are located on the *dha* operon, and there are *gdrA* and *gdrB* sequences at both ends that encode GdrA and GdrB (the activating proteins of GDHt), respectively. Multiple key gene clusters for glycerol metabolism in *Klebsiella pneumoniae* (coenzyme B_12_-dependent) are found in the *dha* operon region [9]. In this operon, *gldA*, *gldB*, and *gldC* in the *dhaB* region encode three subunits of GDHt; *gdrA* and *gdrB* encode the reactivators of GDHt; and *dhaT* encodes 1, 3-propanediol dehydrogenase. When *glpR*, the glycerol inhibitor of GDHt, was knocked out in another *K. pneumoniae* strain, the transformed glycerol production increased by 8.4-fold. Although some studies have improved the conversion efficiency of glycerol via augmenting GDHt in different ways, the systematic feedback inhibition mechanism of glycerol on 3-HPA synthesis in *L. reuteri* remains unclear.

To elucidate the mechanism by which glycerol inhibits 3-HPA synthesis in *L. reuteri* and provide reference data for constructing an *L. reuteri* culture system for highly effective 3-HPA synthesis, we used transcriptome sequencing technology to compare the morphologies and transcriptomes of *L. reuteri* cultured in media with or without glycerol.

## 2. Materials and Methods

### 2.1. Isolation, Morphological Observation, and Antibacterial Activity Analysis of L. reuteri

*L. reuteri* LR301 was isolated from the intestinal microbiota of a healthy mouse and preserved at the School of Life Sciences, Hefei Normal University.

De Man, Rogosa, and Sharpe (MRS) broths containing 0, 400, 600, and 800 mM of glucose were used for the anaerobic propagation of cultures [10]. After incubation at 37 °C for 48 h, the fermentation solution was extracted and centrifuged at 12,000× *g* for 5 min to obtain the supernatant. The antibacterial activity of *L. reuteri* was detected using the inhibition zone test for *Salmonella typhi*, a common conditional pathogen found in pig breeding, as previously described [11].

### 2.2. Heterologous Expression of GDHt

After *L. reukii* was cultured at 37 °C for 24 h, the whole genome of *L. reukii* was extracted using a bacterial DNA extraction kit (AxyPrep, Hangzhou, China). PCR primers Gt30F (5′-CGCCATATGAAACGCCAGAAAC-3′; *Nde*I recognition sequence was underlined) and Gt30R (5′-CCGCTCGAGTTACAGTTCCAGATGT-3′; *Xho*I recognition sequence was underlined) were developed to amplify the *GDHt* gene using genomic DNA as a template. Two restriction endonuclease digestion sites, *Nde*I and *Xho*I, were introduced during PCR. The PCR amplification reaction conditions were initial denaturation at 95 °C for 2 min, followed by 30 cycles of 95 °C for 20 s, 57 °C for 20 s, and 72 °C for 1.5 min, with a final extension at 72 °C for 10 min. Ten microliters of the product was detected using 1% agarose gel electrophoresis. The products were purified to obtain the *GDHt* gene using a DNA in-gel recovery kit (AxyPrep, Hangzhou, China). The purified DNA was inserted between the *Nde*I and *Xho*I restriction sites of plasmid pET28a vector under the action of T4 ligase at 16 °C to generate a recombinant *GDH*t expression plasmid.

The recombinant expression plasmid pET28a-*GDHt* was transformed into *Escherichia coli* BL21 (DE3) to obtain the *GDHt* recombinant expression strain *E. coli* BL21(DE3)-pET28a-*GDHt*. Positive clones were inoculated in test tubes containing 10 mL of liquid lysogeny broth (LB) medium with 100 mg/L of kanamycin and cultured at 37 °C alongside 200 rpm rotary agitation overnight. The final bacterial culture medium was stored at −80 °C.

The recombinant expression strain of *GDHt* was inoculated into shaker flasks containing 100 mL of liquid LB medium at 37 °C with 220 rpm rotary agitation. When the OD600 value of the strain reached 0.8, isopropylthio-β-galactoside (IPTG) inducer, at a final concentration of 0.2 mM, was added, and the medium was cooled to 28 °C to induce expression.

*GDHt* activity in the fermentation broth was detected using a 3-methyl-2-benzothiazolinone hydrazone hydrochloride (MBTH) enzyme activity assay. Each 0.5 mL MBTH enzyme activity assay system contained 0.2 mmol/L of glycerol, 15 μmol/L of vitamin B12, 0.035 mol/L of potassium phosphate buffer at pH 8.0, 0.05 mol/L of KCl, and 100 μL of enzyme solution. After incubation at 37 °C for 10 min, 1.0 mL of 0.1 mol/L potassium citrate solution at pH 3.6 was added to terminate the reaction. Subsequently, 50 μL of 1% MBTH (*w*/*v*) was added and incubated at 37 °C for 15 min. The optical absorbance of the reaction mixture was measured at a wavelength of 305 nm, and enzyme activity was calculated. One unit (U) of enzyme activity was defined as the amount of enzyme required to convert 1 μM of substrate in one minute.

After induction, 2 mL of fermentation broth from each sample was extracted every 2 h and centrifuged at 12,000× *g* at 4 °C for 5 min to collect bacterial cells. The bacterial cells were then ultrasonically lysed to detect enzyme activity. Fermentation was terminated when *GDHt* activity decreased.

### 2.3. Extraction and Purification of GDHt and Evaluation of Effect of Glycerol Concentration on GDHt Activity

The fermentation broth was collected and centrifuged at 12,000× *g* and 4 °C for 20 min to collect the precipitated bacteria. The precipitate was resuspended and cleaned twice using Na-K phosphate-buffered saline (8.0 g/L NaCl, 0.2 g/L KCl, 1.44 g/L Na_2_HPO_4_, and KH_2_PO_4_) at pH 7.0. Then, the cells were ultrasonically lysed and centrifuged at 12,000× *g* at 4 °C for 20 min to obtain 1 L of crude *GDHt* extract from the supernatant. Subsequently, the crude *GDHt* extract was concentrated to 30 mL crude *GDHt* extract using a 300 mL ultrafiltration device with 0.3 MPa of pressure at 4 °C and purified using Ni-column affinity chromatography (details provided in Appendix A).

Different concentrations of glycerol (200, 400, 500, 600, 700, 800, and 1000 mM) were added to the *GDHt* activity assay system at 37 °C to detect the remaining enzyme activity as per the method described above.

### 2.4. Transcriptome Sequencing and Sequence Data Analysis

To analyze the effect of glycerol on *L. reuteri* LR301 at the transcription level, a treatment group (T) supplemented with 600 mmol/L of glycerol and a control group (C) that was not supplemented with glycerol, both of which were contained in culture media, were established, for which there were five replicates. The medium was modified with MRS broth as described previously [12]. *L. reuteri* LR301 cells were anaerobically incubated at 37 °C for 10 h and then centrifuged at 12,000× *g* at 4 °C for 20 min to collect bacteria for transcriptome sequencing. Total bacterial RNA was extracted using TRIzol reagent (Life Technologies, Grand Island, NY, USA) and purified using an RNeasy Micro Kit (QIAGEN, Hiliden, Germany). RNA concentration was measured using a UV spectrophotometer (Mapada, Shanghai, China). First- and second-strand cDNAs were synthesized using a superscript double-stranded cDNA synthesis kit (Life Technologies, Grand Island, NY, USA); then, end repair and 3′ end adenylation were conducted (Biotechnology, Shanghai, China). The synthesized cDNAs were purified using an Agencourt AMPure XP kit (Beckman Coulter, Indianapolis, IN, USA) and ligated to adaptors (Biotechnology, Shanghai, China). Then, the cDNA was enriched, and its concentration was determined using a Qubit 2.0 fluorometer and a Qubit dsDNA HS kit (Life Technologies). Finally, the cDNAs were sequenced on an Illumina HiSeq 2500 platform, according to the manufacturer’s instructions.

Raw reads were trimmed using Trimmomatic v0.36 [13] to remove “N” bases, adaptor sequences, bases with Q-value < 20, reads with length < 35 nt, and their paired reads. The remaining reads were mapped to the reference genome using Bowtie2 2.3.2 [14]. Redundant sequences and insert distributions were analyzed using RSeQC v2.6.1 [15] in accordance with the mapping results. The homogeneity and distribution of the genomic structures were analyzed using Qualimap v2.2.1 [16]. Gene coverage and distribution across the genome were analyzed using BEDTools v2.26.0 [17].

### 2.5. Data Analysis

Data are presented as means ± standard error for each group. Differentially transcribed genes (log_2_ fold change ≥ 1 and *p* < 0.05) were screened using the R DESeq2 package [18], and volcano plots were developed using the R ggplot2 package. Principal co-ordinates analysis (PCoA) and analysis of similarities (ANOSIM) were conducted using the R vegan and vegan3d packages. Results were considered statistically significant at *p* < 0.05.

## 3. Results

### 3.1. Effects of Glycerol on Morphology, Antimicrobial Properties, and GDHt Activity of L. reuteri LR301

Using scanning electron microscopy (SEM), it was observed that *L. reuteri* LR301 was rod-shaped under normal culture conditions (Figure 1A). When the medium contained 600 mM of glycerol, *L. reuteri* LR301 appeared as short rods or spheroids (Figure 1B). Inhibition zone tests revealed that the diameter of the inhibition zone was greatest when the glycerol concentration in the medium was 600 mM. The antibacterial effect weakened when the glycerol concentration was >600 mM (Figure 1C).

SDS-PAGE analysis revealed that the constructed *GDHt* heterologously expressing cells could express GDHt (Figure 1D). MBTH enzyme activity assays showed that within 10 h of induction, GDHt activity increased with fermentation time. The highest enzyme activity level was 356 U/L 10 h after induction (Figure 1E). Moreover, GDHt activity reached a maximum when the glycerol concentration was 600 mM. GDHt activity gradually decreased with an increasing glycerol concentration when the glycerol concentration was >600 mM (Figure 1F).

### 3.2. Effect of Glycerol on L. reuteri Transcriptome

The PCoA profile showed that treatment with 600 mM of glycerol significantly altered the transcriptome of *L. reuteri* LR301 (ANOSIM, R = 1.00, *p* = 0.007; Figure 2A). A heatmap based on the Pearson correlation coefficient also exhibited significant changes in the transcriptome of *L. reuteri* LR301 treated with glycerol (Figure 2B).

To elucidate the effects of glycerol treatment on the transcriptome of *L. reuteri* LR301, differentially transcribed genes were screened using the R DESeq2 package. The results showed that the transcription of 32 genes was significantly up-regulated after glycerol treatment (log_2_ fold change ≥ 1 and *p* < 0.05), including transfer and messenger RNAs, glycoside hydrolase family 31 protein, VIT family protein, α-glucosidase, CBS domain-containing protein, pyrimidine-specific ribonucleoside hydrolase *RihA*, *Lacl* family transcriptional regulator, sensor histidine kinase, NAD(P)H-binding protein, metallophosphoesterase, FAD-dependent oxidoreductase, acyl carrier protein, *YibE/F* family protein, and some tRNAs and hypothetical proteins. Additionally, the transcription of 36 genes was significantly down-regulated after glycerol treatment (log_2_ fold change ≥ 1 and *p* < 0.05), including L-arabinose isomerase, molecular chaperone DnaK, sugar porter family MFS transporter, molecular chaperone *DnaJ*, FGGY-family carbohydrate kinase, SLC45 family MFS transporter, ATP-dependent Clp endopeptidase proteolytic subunit *ClpP*, phosphate ABC transporter permease *PstA*, acetolactate synthase *AlsS*, L-ribulose-5-phosphate 4-epimerase, acetolactate decarboxylase, phosphate ABC transporter ATP-binding protein *PstB*, alpha/beta hydrolase, HAMP domain-containing histidine kinase, lipocalin/fatty-acid-binding family protein, heat-inducible transcriptional repressor *HrcA*, phosphate signaling complex protein *PhoU*, LysM peptidoglycan-binding domain-containing protein, and some tRNAs (Figure 2C).

It is worth noting that although glycerol dehydration is catalyzed by GDHt to form 3-HPA in *L. reuteri* [19,20,21,22], our results showed that the transcriptional level of the *GDHt* gene (*gldA*) was significantly reduced in the *L. reuteri* LR301 cells treated with 600 mM of glycerol compared with the control group (*p* < 0.05). The transcription levels of multiple genes involved in glycol metabolism, including *gldA*, *dhaT*, *glpK*, *plsX*, and *plsY*, were significantly decreased (*p* < 0.05; Figure 3). Kyoto Encyclopedia of Genes and Genomes (KEGG) pathway analysis revealed that these significantly differentially expressed genes were mainly involved in galactose metabolism (Figure 4); purine metabolism (Figure 5); monobactam biosynthesis (Appendix A); nicotinate, nicotinamide (Appendix A), and butanoate metabolism (Appendix A); riboflavin metabolism (Appendix A); and starch and sucrose metabolism (Appendix A). In particular, the transcription levels of genes involved in galactose metabolism, purine metabolism, monobactam biosynthesis, nicotinate and nicotinamide metabolism, and butanoate metabolism were significantly downregulated in the *L. reuteri* specimens treated with 600 mM of glycerol (*p* < 0.05), whereas those of genes involved in D-glucose synthesis in starch and sucrose metabolism were significantly upregulated (*p* < 0.05; Appendix A). Moreover, genes involved in the conversion of sucrose to D-fructose and D-glucose were significantly upregulated (*p* < 0.05; Figure 4), which may lead to the accumulation of D-glucose in *L. reuteri* LR301 cells. Furthermore, the transcription of upstream genes involved in riboflavin metabolism was significantly upregulated in the *L. reuteri* LR301 cells treated with 600 mM of glycerol, whereas those of downstream genes were significantly downregulated (*p* < 0.05; Appendix A). The processes indicated by these results may lead to the intracellular accumulation of 6,7-dimethyl-8-nbityllumazine or riboflavin in *L. reuteri* LR301 cells.

## 4. Discussion

Reuterin can be produced in relatively large amounts via glycerol dehydration catalyzed by GDHt in *L. reuteri* [19,20,21,22], is active against prokaryotic and eukaryotic organisms, and is resistant to proteases and lipases. Therefore, reuterin has broad application prospects in industry, agriculture, food, and other fields [20,23,24]. Reuterin production is affected by oxygen concentrations, glycerin and glucose concentrations, and other fermentation conditions such as temperature, pH, fermentation time, cell age, and biomass concentration [22]. Glycerin concentration is the most important factor affecting reuterin production [22]. GDHt can be produced when *L. reuteri* strains are grown anaerobically on glycerol [25]. Talarico et al. [26] reported a reaction system containing higher concentrations (10 mg/mL) of *L. reuteri* cells incubated anaerobically in the presence of 250 mM of glycerol that produced reuterin at concentrations as high as 1000 minimum inhibitory concentration (MIC) units/mL. Sun et al. [22] reported that a maximum concentration of reuterin could be obtained by using 25 g/L of 20 h old *L. reuteri* DPC6 cells to ferment 350 mM of glycerol for 2 h at 25 °C and pH 6.8. Our results indicated that when the glycerol concentration in the fermentation broth exceeded 600 mM, GDHt activity decreased with the increasing glycerol concentration, which might have been due to the inactivation of GDHt when the glycerol concentration in the substrate was too high.

The genes encoding GDHt are located on the *dha* operon, and there are *gdr*A and *gdr*B sequences at both ends that encode GdrA and GdrB (the activating proteins of GDHt), respectively. Multiple key gene clusters for glycerol metabolism in *K. pneumoniae* (co-enzyme B_12_-dependent) have been found in the *dha* operon region [9,27]. In this operon, *gld*A, *gld*B, and *gld*C in the *dha*B region encode three subunits of GDHt; *gdr*A and *gdr*B encode the reactivators of GDHt; and *dha*T encodes 1, 3-propanediol dehydrogenase. When *glpR*, the glycerol inhibitor of GDHt, was knocked out in another *K. pneumoniae* strain, the transformed glycerol production increased 8.4-fold. However, our results showed that the transcription levels of the *gldA* and *dhaT* genes in the *L. reuteri* cells cultured in the medium containing 600 mM of glycerol decreased significantly compared to those of the control (Figure 3). Since we did not design the medium for adding glycerol at other concentrations, we could not analyze the glycerol content at different concentrations in the fermentation broth, which would have allowed us to determine how this consideration might influence the trends in the transcriptional levels of the *glaA* and *dhaT* genes in the *L. reuteri* cells. Our results suggest that the regulation of reuterin synthesis may not only depend on GDHt and 1, 3-propanediol dehydrogenase but may also involve a more complex regulatory process that needs to be further clarified.

Lin et al. [12] identified a putative genomic island containing 12 genes (*rtcP*, *rtcK*, *rtcN*, *rtcA*, *rtcB*, *rtcC*, *rtcR*, *rtcS*, *rtcT,* and 3 other genes coding for components of ABC transporters) responsible for reutericyclin biosynthesis using a comparative genomic approach. Greppi et al. [20] reported that an *L. reuteri* strain isolated from a chicken gastrointestinal tract harboring *pdu-cob-cbi-hem* genes produced between 156 ± 11 and 330 ± 14 mM 3-HPA. Moreover, all erythromycin-resistant strains carried the *ermB* gene. However, we did not detect the expression of these genes. This was probably due to the different annotation information of the reference genome.

Reuterin production is inhibited by excess glucose present in a glycerol solution because the NAD+ produced during the conversion of glycerol to reuterin can be used in glucose metabolism [22]. Reuterin can be further converted to 1,3-propanediol (1,3-PD) by propanediol dehydrogenase, depending on the NAD+ concentration. Thus, the amount and characteristics of reuterin produced by *L. reuteri* depend on the glucose/glycerol molar ratio [22]. The ratio required for maximum reuterin production is 0.33 [28]. Our results showed that adding 600 mM of glycerol to the fermentation broth increased the transcriptional levels of genes involved in D-glucose production in *L. reuteri* (Appendix A), indicating that there may not be sufficient D-glucose for *L. reuteri* growth and reuterin synthesis. Therefore, the addition of glucose to fermentation broth may further increase reuterin production. If this inference is confirmed by further experiments, this indicates that the analysis of gene transcription levels via transcriptome sequencing can provide guidance for optimizing microbial fermentation conditions.

It is worth noting that since we only compared the transcriptomes of *L. reuteri* with and with the addition of 600 mM of glycerol, the effect of other concentrations of glycerol on the *L. reuteri* transcriptome should be considered in subsequent studies. Moreover, the future studies should consider the correlation between the transcriptome and fermentation and the metabolic characteristics (such as glycol consumption and 3-HPA production) of *L. reuteri* under different glycerol concentrations, as such a consideration would be crucial to clarifying the mechanism of glycerol concentrations affecting reuterin production in *L. reuteri*.

## 5. Conclusions

The addition of 600 mM of glycerol to the *L. reuteri* LR301 culture medium and incubation for 10 h at 37 °C resulted in the best bacteriostatic effect, and the morphology of *L. reuteri* changed significantly. The addition of 600 mM of glycerol to the culture medium significantly changed the transcriptome and significantly downregulated the transcription of genes involved in glycol metabolism, such as *gldA*, *dhaT*, *glpK*, *plsX*, and *plsY*, but significantly upregulated the transcription of genes related to D-glucose synthesis. These results imply that high-concentration glycerol likely down-regulates reuterin production through down-regulating glycolipid metabolism, galactose metabolism, and purine metabolism in *L. reuteri*. Although transcriptome analysis alone cannot sufficiently explain the inhibitory mechanism of glycerol on reuterin synthesis in *L. reuteri*, our study provides reference data for constructing an *L. reuteri* culture system for highly effective reuterin synthesis. Further experimental verification of our results using genetic engineering methods is necessary to provide important experimental data support for constructing an efficient reuterin synthesis system.

## Figures and Tables

**Figure 1 microorganisms-11-02007-f001:**
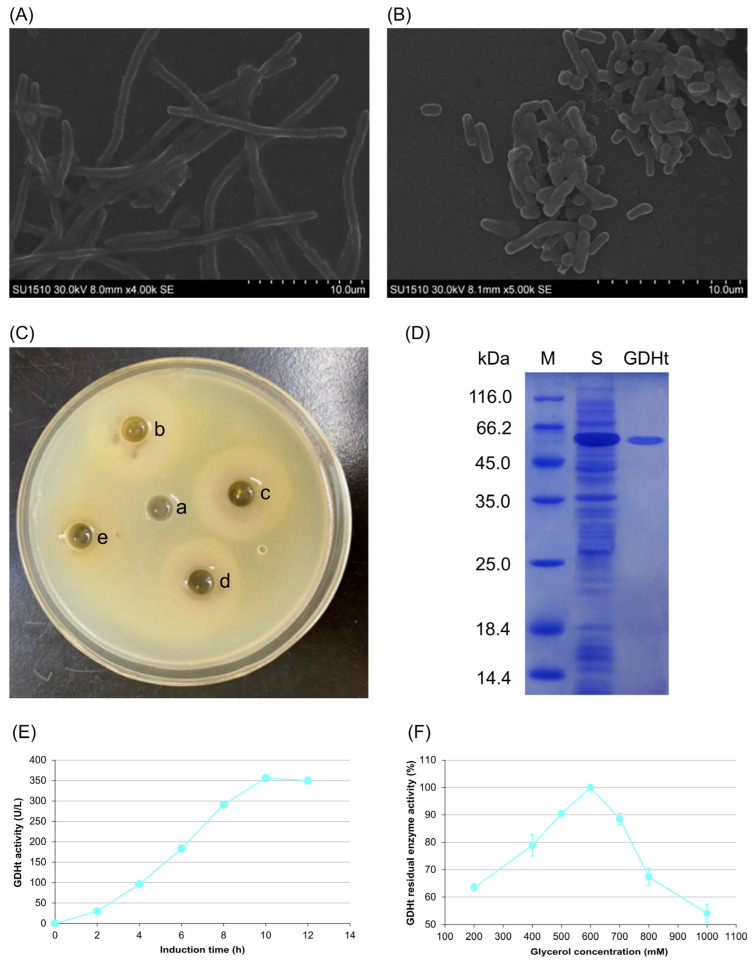
Effects of glycerol on morphology, bacteriostasis, and GDHt activity of *L. reuteri LR301*. Electron microscopy images of *L. reuteri* in control (**A**) and treatment (**B**) groups. (**C**) Results of inhibition zone test: a, 0 mM of glycerol; b, 200 mM of glycerol; c, 400 mM of glycerol; d, 600 mM of glycerol; e, 800 mM of glycerol. (**D**) SDS-PAGE profile of GDHt. M denotes the marker, S is the crude enzyme after concentration and ultrasonic fragmentation, and GDHt is the purified protein band of GDHt. (**E**) Changes in GDHt activity of *L. reuteri* with fermentation times. (**F**) Changes in GDHt activity of *L. reuteri* with glycerol concentrations. GDHt activity in the fermentation broth was detected using a 3-methyl-2-benzothiazolinone hydrazone hydrochloride enzyme activity assay with 0.2 mmol/L of glycerol, 15 μmol/L of vitamin B12, 0.035 mol/L of potassium phosphate buffer at pH 8.0, 0.05 mol/L of KCl, and 100 μL of enzyme solution.

**Figure 2 microorganisms-11-02007-f002:**
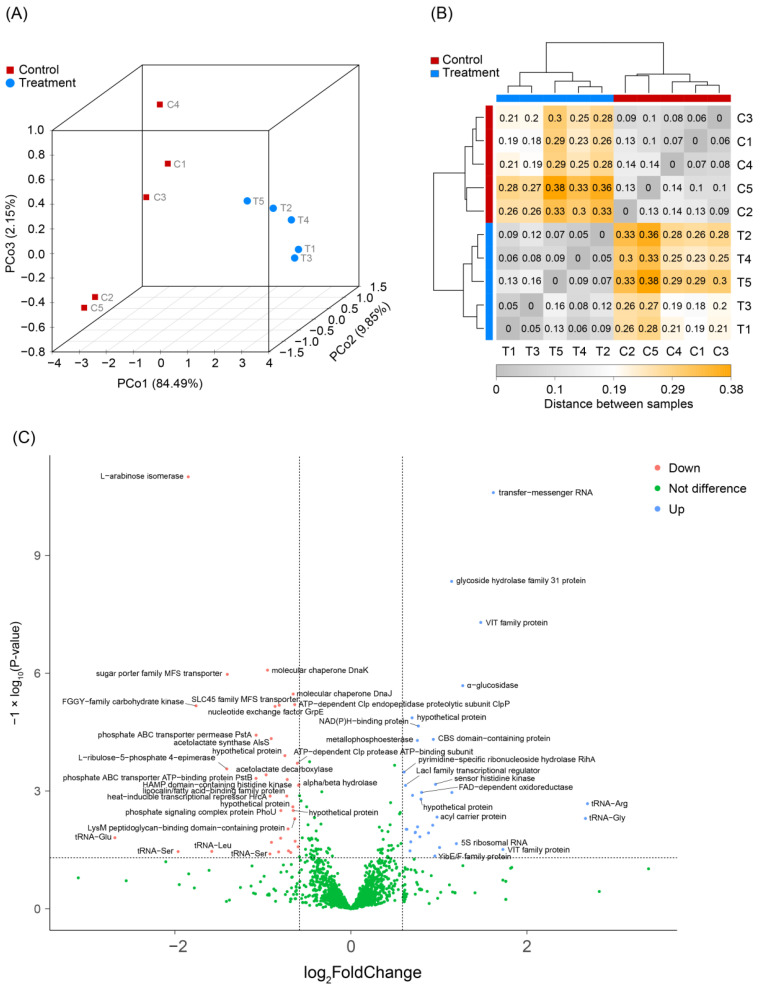
Effect of 600 mM of glycerol on the transcriptome of *L. reuteri*. (**A**) PCoA. (**B**) Heatmap. (**C**) Volcano plot showing genes with significant differences (log_2_ fold change ≥ 1 and *p* < 0.05) in transcription in *L. reuteri* treated with glycerol.

**Figure 3 microorganisms-11-02007-f003:**
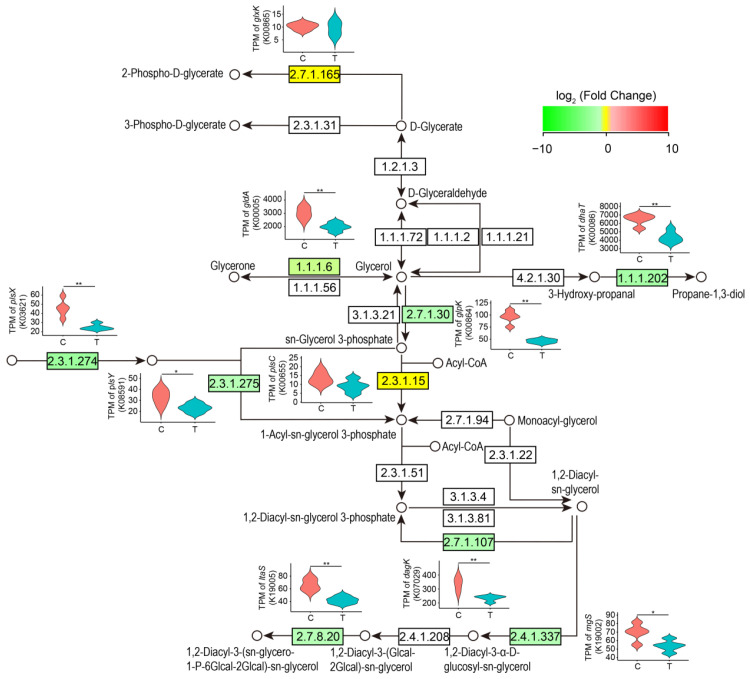
Altered genes participating in glycerolipid metabolism in *L. reuteri* treated with 600 mM of glycerol. Green and yellow indicate that the transcription levels of genes were significantly down-regulated (*p* < 0.05) and not significantly changed (*p* ≥ 0.05), respectively. C, *L. reuteri* treated without glycerol; T, *L. reuteri* treated with 600 mM of glycerol. * *p* < 0.05; ** *p* < 0.01.

**Figure 4 microorganisms-11-02007-f004:**
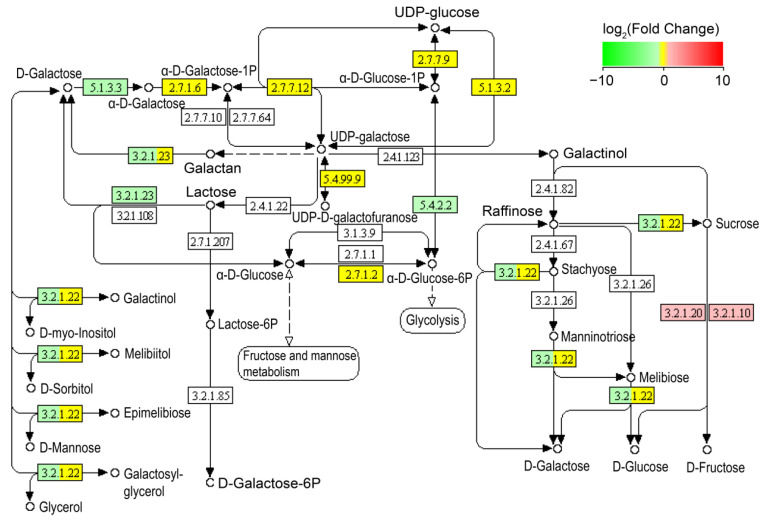
Altered genes participating in galactose metabolism in *L. reuteri* treated with 600 mM of glycerol. Red, green, and yellow indicate that the transcription levels of genes were significantly up-regulated (*p* < 0.05), down-regulated (*p* < 0.05), and not significantly changed (*p* ≥ 0.05), respectively.

**Figure 5 microorganisms-11-02007-f005:**
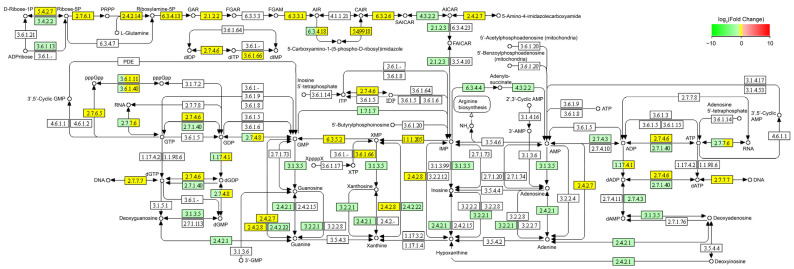
Altered genes participating in purine metabolism in *L. reuteri* treated with 600 mM of glycerol. Green and yellow indicate that the transcription levels of genes were significantly down-regulated (*p* < 0.05) and not significantly changed (*p* ≥ 0.05), respectively.

## Data Availability

The data that support the findings of this study are openly available in [NCBI SRA] at [https://www.ncbi.nlm.nih.gov/sra, 23 June 2023], accession number: PRJNA936277.

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
