# Peer review of "Transcriptome Analysis of Glycerin Regulating Reuterin Production of Lactobacillus reuteri"

_microorganisms, 2023, doi:10.3390/microorganisms11082007_

Round 1
Reviewer 1 Report (New Reviewer)
Please try to use in the final Ms version electron micrographs of better quality
I suggested minor editing in English, particularly because of the lack of use of the conditional tense (see discussion, lines 318-320)
Author Response
Comments and Suggestions for Authors
Please try to use in the final Ms version electron micrographs of better quality
Responses
Thank you for your comment. We have revised the manuscript using in the final Ms version electron micrographs according to your comment.
Comments on the Quality of English Language
I suggested minor editing in English, particularly because of the lack of use of the conditional tense (see discussion, lines 318-320)
Response
We have revised our manuscript according to your comment.
Reviewer 2 Report (New Reviewer)
The manuscript entitled " transcriptome analysis of glycerine regulating reuterin production of Lactobacillus reuteri" described in depth the regulation and inhibition of reuterin production in presence of glycerol.
The authors described the regulation of some genes in different metabolic pathways. The results showed a down regulation of genes of glycerol metabolism and up regulation of genes involving D-glucose synthesis.
The manuscript is clear, but there are some questions.
1- Regarding the transcriptome analysis, which housekeeping gene the authors used to stablish the up-regulation or down-regulation of the genes in the diferent conditions of growth?
2.-Page 10 line 277. Why authors used mg/ml to described concentration of L.reuteri cells, instead of CFU/ml, did you know the weight of L.reuteri?. Is it dry weight?
Author Response
Comment
1- Regarding the transcriptome analysis, which housekeeping gene the authors used to stablish the up-regulation or down-regulation of the genes in the different conditions of growth?
Response
We did not use the housekeeping gene method to analyze the transcriptome, but used the RPKM value method to standardize the transcriptome data to eliminate the impact of sequencing depth and gene sequence length on the transcriptional result. And then we used the R DESeq2 package to screen differential transcriptional genes according to the standard of log2 fold change ≥ 1 and p < 0.05, which is a common analysis method for RNA-Seq. We have supplemented the description of the analysis process in the Materials and Methods section of our manuscript.
Comment
2.-Page 10 line 277. Why authors used mg/ml to described concentration of L.reuteri cells, instead of CFU/ml, did you know the weight of L.reuteri?. Is it dry weight?
Response
Thank you for your comment. According to the description of Talarico et al. (1998), this is the suspended concentration of dry weight of L. reuteri cells.
Reviewer 3 Report (Previous Reviewer 1)
Please see the attached file for review comments.

Please see comments no. 2, 4 and 6-8 of the review report.
Author Response
Comment
- Two new scientists (Jiajia Ni and Zhongdong Xu) who were not listed in the original version of the manuscript (“Molecular mechanism of glycerin regulating bacteriostasis of Lactobacillus reuteri”) were included among the authors of the new version of the article (“Transcriptome analysis of glycerin regulating reuterin production of Lactobacillus reuteri”). However, their contribution has not been described, and no new research has been reported in the new version of the article. This very important issue must be addressed.
Response
Thank you very much for your comment. Jiajia Ni had assisted in the analysis of data as our description in the Acknowledgments section in the initial draft of our manuscript. We believe that only assisting in the analysis of data did not achieve the level of contribution as a co-author. However, in the subsequent process of the manuscript revision, Jiajia Ni made substantial contributions to the modification of the manuscript, so we listed him as a co-author. Moreover, as Jiajia Ni is an employee of Guangdong Meilikang Bio-Science Ltd., China, we have revised the Conflicts of Interest section. Zhongdong Xu contributed with the data analysis and final revision of the manuscript. We have added the description in the Author Contributions section of our revised manuscript.
Comment
- Line 14: Reviewer comment no. 1 of the previous review report has not been fully taken into account in the new version of the manuscript. The statement “Reuterin can be produced from glycerol dehydration catalyzed by glycerol dehydratase” should be corrected. I suggest, for example, the phrase "Reuterin can be produced from glycerol in a dehydration reaction catalyzed by glycerol dehydratase” or "Reuterin can be produced from glycerol in an enzymatic reaction catalyzed by glycerol dehydratase”.
Response
Thank you very much for you comment. We have revised the sentence according to your comment.
Comment
- Line 28: Reviewer comment no. 3 of the previous review report has not been fully taken into account in the new version of the manuscript. In Keywords it should be “3-hydroxypropionaldehyde” instead of “3-hydroxypropannaldehyde”.
Response
Thank you very much for your comment. We have revised the keyword according to your comment.
Comment
- Line 43: Reviewer comments no. 5 and 9 of the previous review report have not been fully taken into account in the new version of the manuscript. It should be “coenzyme B12” instead of “co-enzyme B12”.
Response
Thank you for your comment. We have revised the word according to your comment.
Comment
- Lines 200-201, Figure 1 caption: I suggest “using a 3-methyl-2-benzothiazolinone hydrazone hydrochloride (MBTH) enzyme activity assay” instead of “using a 3-methyl-2-benzothiazolinone hydrazone hydrochloride enzyme activity assay”. See lines 120-121 for comparison.
Response
Thank you for your comment. We have revised the caption according to your comment.
Comment
- Line 268: Reviewer comment no. 29 of the previous review report has not been fully taken into account in the new version of the manuscript. The statement “Reuterin can be produced from glycerol dehydration catalyzed by glycerol dehydratase” should be corrected. I suggest, for example, the phrase "Reuterin can be produced from glycerol in a dehydration reaction catalyzed by glycerol dehydratase” or "Reuterin can be produced from glycerol in an enzymatic reaction catalyzed by glycerol dehydratase”. See also comment no. 2.
Response
Thank you very much for your comment. We have revised the sentence according to your suggestion.
Comment
- Lines 322-323: The statement “with and with the addition of 600 mM glycerol” should be corrected.
Response
Thank you very much for your comment. It should be “with and without the addition of 600 mM glycerol”. We have revised the error.
Comment
- Lines 324-328: The sentence is not easy to understand. I suggest editing of English language.
Response
Thank you for your comment. We have revised the sentences according to your comment.
Comment
- Line 353: It should be “Wei Wang, Ran Zhang” instead of “Wei Wang Ran Zhang”.
Response
Thank you for your comment. We have revised our manuscript according to your comment.
Comment
- Lines 352-355, Author Contributions: The contribution of two authors of the previous and resubmitted version of the manuscript (Han Bai and Minghui Zhou) is not described. This important issue should be addressed.
Response
Han Bai and Minghui Zhou performed the experiments with other authors. We have revised the Author Contributions section in our revised manuscript.
Comment
- Author contributions: See comment no.1 for contribution of two new scientists (Jiajia Ni and Zhongdong Xu) not included among the authors of the previous version of the manuscript.
Response
Thank you very much for your comment. Jiajia Ni had assisted in the analysis of data as our description in the Acknowledgments section in the initial draft of our manuscript. We believe that only assisting in the analysis of data did not achieve the level of contribution as a co-author. However, in the subsequent process of the manuscript revision, Jiajia Ni made substantial contributions to the modification of the manuscript, so we listed him as a co-author. Moreover, as Jiajia Ni is an employee of Guangdong Meilikang Bio-Science Ltd., China, we have revised the Conflicts of Interest section. Zhongdong Xu contributed with the data analysis and final revision of the manuscript. We have added the description in the Author Contributions section of our revised manuscript.
Comment
- Acknowledgments: Guangdong Meilikang Bio-Science Ltd., China is indicated as affiliation of Jiajia Ni (one of newly added authors). Co-authorship or acknowledgments? This issue should be addressed. See also comments no. 1 and 11.
Response
Thank you very much for your comment. Jiajia Ni had assisted in the analysis of data as our description in the Acknowledgments section in the initial draft of our manuscript. We believe that only assisting in the analysis of data did not achieve the level of contribution as a co-author. However, in the subsequent process of the manuscript revision, Jiajia Ni made substantial contributions to the modification of the manuscript, so we listed him as a co-author. Since Jiajia Ni was listed as a co-author, there is no longer a need for the acknowledgement section, so we have deleted it. Moreover, as Jiajia Ni is an employee of Guangdong Meilikang Bio-Science Ltd., China, we have revised the Conflicts of Interest section.
Round 2
Reviewer 3 Report (Previous Reviewer 1)
Please see the attached file for review comments.

Please see comments 1-3 in the review report.
Author Response
Comment
- Line 43: Reviewer comment no. 4 has not been taken into account in the revised version of the manuscript. It should be “coenzyme B12” instead of “co-enzyme B12”. The authors' answer to comment no. 4 is not true.
Response
Thank you for your comment. We had changed the word according to your comment. However, due to our use of Microsoft Word’s change tracking function to display modified content, the connectionstring “-“ overlapped with delete symbol, causing you to mistakenly believe that we have not modified the word.
Comment
- Lines 326-327: I suggest “the effect of other concentrations of glycerol on L. reuteri transcriptome should be further studied.” instead of “the effect of other concentrations of glycerol on L. reuteri transcriptome should be further study.”
Response
Thank you very much for your comment. “Be further studied” is a passive tone. There will be syntax error if the sentence is changed according to your comment, so we did not modify it.
Comment
- Line 330: I suggest “also should be further studied” instead of “also should be further study”.
Response
Thank you very much for your comment. “Be further studied” is a passive tone. There will be syntax error if the sentence is changed according to your comment, so we did not modify it.
Comment
- Final comment: I am a bit surprised that the authors did not treat the statement regarding their contribution in a responsible manner. It should be noted that there is a statement in the each version of the article "All authors revised and approved the final version of the manuscript.”
Response
Thank you very much for your comment. We had carefully revised our manuscript and obtained confirmation from all authors. As a result, and due to the summer vacation, we did not complete the confirmation within the 10 days and delayed the revision of our manuscript. We had already revised the author contributions statement in the Author Contribution section of our manuscript, but due to our inability to modify the author contributions statement in the submission system at this stage, the author contribution statement in the submission system is still in the initial state. We will contact the editor to modify the author contributions statement.
This manuscript is a resubmission of an earlier submission. The following is a list of the peer review reports and author responses from that submission.
Round 1
Reviewer 1 Report
Please see the attached file for comments.

Reviewer 2 Report
The present manuscript aimed to elucidate the molecular mechanism regulated by glycerin on the synthesis of 3-HPA in the bacteria L. reuteri. The authors carried out assays evaluating the effect of glycerol supplementation on the cell morphology of L. reuteri, bacteriostatic effect of reuterin, glycerol dehydratase activity and transcriptome profile of L. reuteri. This issue is very important for the industrial and healthy application. However, the set of experiments and the results introduced and discussed by the authors did not answer the main goal of the manuscript. In addition, no molecular mechanism was proposed and the authors did not link the first part of the results (cell morphology, bacteriostatic effect and enzyme activity) to the transcriptome profile. It seems two independent work. The manuscript require a deep revaluation in the interpretation of the data, especially in the inhibition concentration of the glycerol and transcriptome data. No fermentation parameters were quantified, such as glycerol consumption and 3-HPA production, these information are essential to understand the transcriptome analysis. Biological process analysis is also needed to better understand the up and down-regulated genes. Actually, only transcriptome analysis is not enough to explain enzymatic inhibition.
Reviewer 3 Report
Dear Editor,
I inform you that at this moment I am very sick and I am not able to correct your article. I apologize again.